# A mathematical model assuming frequency-dependent cost for analyzing the influence of cell competition on radiation effects

Kouki Uchinomiya[ORCID]*, Masanori Tomita[ORCID]

Biology and Environmental Chemistry Division, Sustainable System Research Laboratory, Central Research Institute of Electric Power Industry, Abiko-shi, Chiba, Japan

* u-kouki@criepi.denken.or.jp

## Abstract

High dose-rate ionizing radiation exposure increases the risk of cancer; however, the health effects of low dose-rate exposure remain controversial. Stem cell competition entails stem cells with lower fitness being eliminated from the stem cell pool when interacting with neighboring stem cells with higher fitness. It is hypothesized that this phenomenon reduces radiation damage under very low dose-rate conditions rather than high dose-rate irradiation conditions. If cell-cell interactions exist, the frequency of cells in a population affects the fitness of cells. However, they have not been adequately considered in discussions of radiation damage accumulation. We developed a mathematical model to analyze the influence of radiation-induced stem cell competition on the accumulation of radiation damages. Two cell types were assumed: intact and damaged. An intact cell can be damaged by a hit of radiation track. Cell division and elimination follow the Moran process, wherein the probability of cell elimination depends on the frequency-dependent cost. Under low dose-rate irradiation conditions, the size of the cell pool can determine whether competition promotes or suppresses the accumulation of damaged cells, even when the cost parameters are the same. In addition, the presence or absence of spatial structure can qualitatively alter the effect of competition on damage accumulation. If cells had a higher cost when interacting with different cell types, the results were qualitatively different compared to scenarios with no spatial structure. Thus, considering the spatial structure is essential for an in-depth understanding of the effects of stem cell competition.

## 1. Introduction

Ionizing radiation is a well-known external factor that increases the risk of cancer [1]. However, the cancer risk associated with low dose and low dose-rate of radiation remains controversial. For example, an epidemiological study conducted in a high

**Data availability statement:** All relevant data are within the manuscript and its Supporting Information files.

**Funding:** KU was supported by JSPS (Japan Society for the Promotion of Science) KAKENHI Grant Number JP 20K19972. The funders had no role in study design, data collection and analysis, decision to publish, or preparation of the manuscript.

**Competing interests:** The authors have declared that no competing interests exist.

natural background radiation area (HBRA) in Kerala, India, showed that no cancer site was significantly associated with the cumulative radiation dose [2,3]. From a radiation protection perspective, the influence of radiation exposure at low dose rates on the risk of carcinogenesis is an important factor. The International Commission on Radiological Protection (ICRP) published papers discussing the health effects of low dose and/or low dose-rate radiation [4–7]. Although the United Nations Scientific Committee on the Effects of Atomic Radiation (UNSCEAR) described that a low dose rate for external low LET radiation is < 0.1 mGy per min (averaged over one hour or less) [8], the biological data under lower dose-rate irradiation conditions are insufficient [7]. Various biological events, such as DNA repair, apoptosis and cellular senescence, are known to be induced for protection against radiation-induced stochastic effects. In addition, the ICRP proposed a hypothesis that competitive elimination of damaged stem cells suppresses the effect at very low dose rates on the order of a few mGy per month or year, which is close to the background radiation levels [1]. When the dose rate is very low, some stem cells may be damaged and mixed with intact stem cells in the stem cell pool. If the damaged cells are removed from the stem cell pool, the effect of radiation is not observed.

Cell competition is a phenomenon in which cells that are more adapted to the environment, that is, cells with higher fitness, exclude cells with lower fitness [9,10]. From a cell biology perspective, cellular fitness can be interpreted as the capacity of a cell to proliferate; cells that divide actively and exhibit lower susceptibility to cell death can therefore be regarded as possessing high fitness. Stem cell competition, a type of cell competition, can result in the competitive elimination of damaged cells. Depending on the mutation type, cell competition can promote or suppress cancer [11]. Especially, the phenomenon of normal cells sensing and actively eliminating neighboring transformed cells in epithelial tissues is referred to as epithelial defense against cancer [12]. The hypothesis that stem cell competition suppresses the effects of radiation has been supported by experiments [13,14]. The proliferation of irradiated cells did not differ from that of non-irradiated cells when cultured or transplanted alone; however, when mixed, the proliferation of irradiated cells was suppressed [13,14]. This indicates that the mixing of irradiated and non-irradiated cells changes the fitness of the cells. Fitness can be classified into two types: frequency-independent fitness and frequency-dependent fitness. When fitness is determined solely by the environment and the properties of cells, frequency-independent fitness is used. Frequency-dependent fitness is used when fitness is determined not only by the nature of the environment and its cells but also by the frequency of other cells present in the environment. Several mathematical models, such as the Lotka-Volterra model [15,16] and the replicator dynamics model [17–19], have been used to analyze cell competition. We have developed a mathematical model that demonstrates the suppression of the effects of low dose-rate radiation by stem cell competition when damaged cells have lower division rate and/or are likely to be eliminated [20]. This mathematical model assumed that proliferation and elimination, which are related to fitness, are independent of the frequency of other cells. However, experimental results showed that the proliferation of irradiated cells is influenced by non-irradiated

cells [14]. Although radiation can affect cell proliferation rates (e.g., [21]), but most cancer risk studies assume the intrinsic proliferative capacity of cells. If we consider fitness to be an indicator of cell proliferation ability, assuming frequency-dependent fitness is crucial for studying the effects of cell competition in more detail.

When discussing frequency-dependent fitness, it is useful to apply evolutionary game theory, which was constructed to analyze the evolution of conflicts between animals of the same species [22]. The concept of this theory has been applied in cancer research [23–25]. A typical application is called a matrix game, in which there are several types of cells, and each pairwise interaction between cell types is assigned a parameter called a payoff [25]. The frequency dynamics of each cell type are mathematically described, and the payoff is measured experimentally. For example, the co-culture of cancer cells changes the parameters of the game [17], and some factors that affect the results of the game are well-known [26–29]. When considering cell competition, spatial structure is important because cell competition can occur through short-range interactions such as cell–cell contact.

In this study, we first constructed a mathematical model to discuss the effects of dose rates with frequency-dependent fitness based on the evolutionary game theory. Next, we calculated an approximate solution of the model for very low dose-rate irradiation conditions and investigated how it varied with the total number of cells in the cell pool. Finally, we modified the model assuming a spatial structure and compared it with the case without a spatial structure.

## 2. Mathematical models

### 2.1 Abstraction of the well-mixed model and lattice space model

First, we considered the well-mixed model. We assumed a stem cell pool containing $N$ cells and two types of stem cells: intact and damaged. Similar to our previous study [20], there are three events: stochastic transition, cell division, and elimination of a cell. The stochastic transition corresponds to radiation exposure. When an intact cell is exposed to radiation, it might convert into a damaged cell. We termed this change as "transition." The assumptions used for cell division and elimination differ from those used in the previous study [20]. In the previous study, cell division was assumed to occur before the elimination of a cell. Both cell division and elimination depend on the intrinsic character of cells. In this study, we assumed that cell division occurred randomly and that the probability of cell elimination depended on frequency-dependent costs. We used the term 'cost' to focus on cell elimination. As the cells with higher costs are more likely to be eliminated from the stem cell pool, higher costs can be interpreted as lower fitness. We also assumed that cell division and elimination occurred simultaneously. These assumptions make the model a Moran process [30], commonly used in evolutionary biology studies. In the lattice space model, the stochastic transition is similar to that in the well-mixed model. The cost of a cell is determined by its type and the types of surrounding cells. The higher the cost of a cell, the higher the probability of its elimination. If a cell is eliminated, its position is replaced by a copy of either itself or one of its neighboring cells, chosen at random.

### 2.2 Well-mixed model

The stochastic transition assumption is almost the same as that used in a previous study [20]. There are $r$ units of radiation, and an intact cell becomes a damaged cell with probability $H$ when a radiation track hits an intact cell. We assumed that the damage is irreversible, and that any effects of repair are implicitly included in $H$. When a radiation track hits a damaged cell, nothing changes. These two parameters are combined into one parameter $\lambda = rH$, which is the expected number of transitions per time step, given a sufficiently large number of intact cells. We have assumed that the "transition" occurs only by radiation. Therefore, $\lambda$ can be assumed as the dose rate. In addition, it was assumed that radiation only affects "transition" and does not affect other parameters. After the stochastic transition, cell division and elimination occur. One cell is randomly selected for division. Let $k$ be the number of damaged cells. Then, the number of intact cells is $N - k$. The probabilities of an intact cell dividing and a damaged cell dividing are $\frac{(N-k)}{N}$ and $\frac{k}{N}$, respectively. Cell

elimination depends on the cost of each cell type. To consider the influence of cell–cell interactions, we assumed that the cost depended not only on the type of cell but also on the types of other cells. Here, we introduced a simple game theory framework. If only cells of the same type are present, there is only one relationship to consider (Fig 1A and 1B). When two types of cells are mixed, there are relationships between different types of cells, in addition to the relationship between cells of the same type. Therefore, it is necessary to consider the four types of relationships when determining the cost (Fig 1C). The influence of cell–cell interactions is summarized in a matrix called the cost matrix (Fig 1D). When one intact cell interacts with the other intact cell, the intact cell incurs cost $C_{I \leftarrow I}$. In another case, when one intact cell interacts with a damaged cell, the intact cell incurs cost $C_{I \leftarrow D}$. The expected cost of an intact cell when there are $k$ damaged cells, $F_k$, is defined as follows:

$$F_k = C_{I \leftarrow I} \frac{N - k - 1}{N - 1} + C_{I \leftarrow D} \frac{k}{N - 1}$$

(1)

If $C_{I \leftarrow I} = C_{I \leftarrow D}$, the cost is frequency-independent. Similarly, when a damaged cell interacts with an intact cell, the damaged cell incurs cost $C_{D \leftarrow I}$. If the damaged cell interacts with another damaged cell, the damaged cell incurs cost $C_{D \leftarrow D}$. Then, the expected cost of a damaged cell when there are $k$ damaged cells, $G_k$ is

$$G_k = C_{D \leftarrow I} \frac{N - k}{N - 1} + C_{D \leftarrow D} \frac{k - 1}{N - 1}$$

(2)

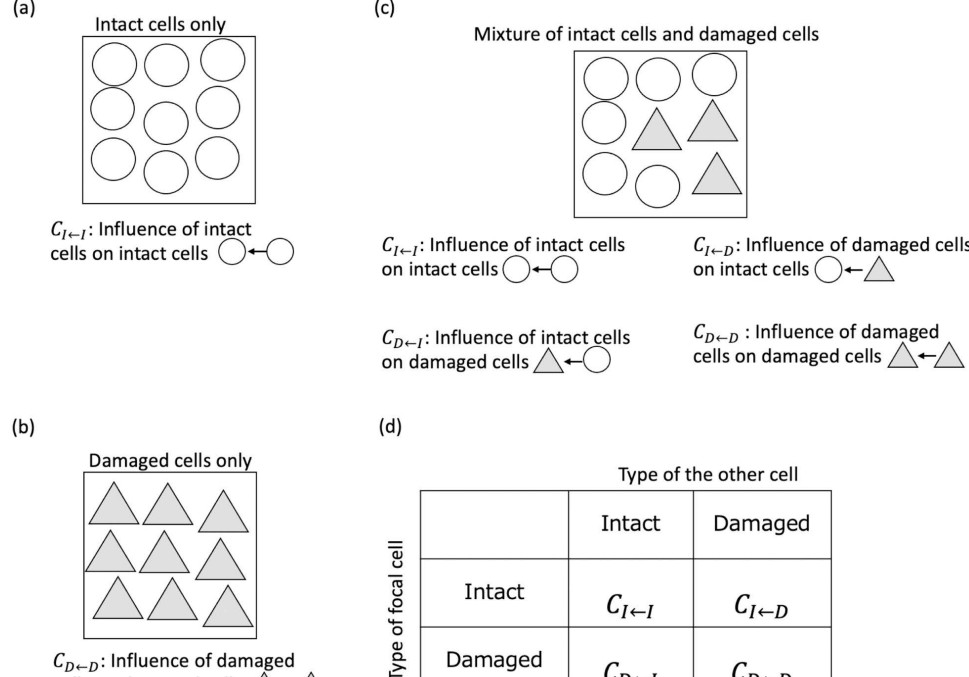

**Fig 1. The abstraction of cell–cell interaction.** (a) (b) If only the same cells are present, there is only one interaction that needs to be considered. (a) When there are only intact cells, only interactions among intact cells are considered. (b) When there are only damaged cells, only interactions among damaged cells are considered. (c) When two different types of cells are mixed, in addition to interactions among the same cells, there are interactions between cells of different types. Therefore, four interactions must be considered. (d) The result of cell–cell interaction is summarized in a cost matrix. For example, when an intact cell interacts with a damaged cell, the intact cell suffers cost $C_{I \leftarrow D}$.

If $C_{D \leftarrow I} = C_{D \leftarrow D}$, the cost is frequency-independent. Then, the probability of elimination of an intact cell and a damaged cell is defined as $(N-k)F_k / \{(N-k)F_k + kG_k\}$ and $kG_k / \{(N-k)F_k + kG_k\}$, respectively. The numerator is the expected cost of each cell type multiplied by the number of cells and the denominator is the total cost of the population.

The change in the number of damaged cells due to cell elimination and division is limited to three patterns. If an intact cell is eliminated and a damaged cell divides, the number of damaged cells increases by one. However, if a damaged cell is eliminated and an intact cell divides, the number of damaged cells decreases by one. In the other cases, the number of damaged cells does not change. Let $P_{i,j}$ be the probability that the number of damaged cells changes from $i$ to $j$ through cell division and elimination. When $k$ damaged cells exist before these processes, $P_{i,j}$ can be summarized as follows:

$$P_{k,k+1} = \frac{k}{N} \frac{(N-k)F_k}{(N-k)F_k + kG_k},$$

(3a)

$$P_{k,k-1} = \frac{N-k}{N} \frac{kG_k}{(N-k)F_k + kG_k},$$

(3b)

$$P_{k,k} = 1 - P_{k,k+1} - P_{k,k-1},$$

(3c)

$$P_{k,j \neq k+1, k-1, k} = 0$$

(3d)

After cell division and elimination, the process returns to stochastic transitions. These processes are repeated until the damaged cells occupy the cell pool. Since one repetition results in one cell division and elimination, $N$ repetitions correspond to the time of the cell cycle, which is used as a unit of time of the model. The well-mixed model is a Markov process, and the unique absorbing state is when the cell pool is entirely occupied by damaged cells, as no external influx of intact cells is assumed (Appendix A). As a criterion for the influence of radiation exposure, we use the average time until damaged cells occupy the cell pool, written as $T_{abs}$. In other words, $T_{abs}$ is the number of cell cycles required for the cell pool to change from entirely intact cells to entirely damaged cells. The reciprocal of this criterion is often used in evolutionary biology to represent the rate of evolution [31]. Without loss of generality, we set $C_{I \leftarrow I} = 1$. This means that the costs are standardized by the cost of interaction between intact cells. For example, if $C_{I \leftarrow D} > 1$, the cost to an intact cell is greater when interacting with a damaged cell than when interacting with another intact cell. Conversely, if $C_{I \leftarrow D} < 1$, the cost is greater when interacting with another intact cell than with a damaged cell.

## 2.3 Lattice space model

In the lattice space model, the cells are arranged in lattice space. Each lattice site is occupied by a single cell, and the total number of lattice sites is finite. The stochastic transition is the same as that in the well-mixed model. In stochastic transition, $r$ unit of radiation exists, and a cell is selected by a single unit of radiation. Subsequently, the transition occurs with probability $H$ when an intact cell is chosen. The differences between the well-mixed model and the lattice space model are cell elimination and cell division. The cell to be eliminated is chosen stochastically depending on the cost. Then, one cell is randomly chosen from the removed cell and surrounding cells to divide and fill the removed space. In other words, clones of the removed cell or its surrounding cells arise in the removed space. The cost of a cell depends on its interaction with surrounding cells based on the cost matrix (Fig 1D). We assumed that a cell interacts with eight surrounding cells. This is the so-called Moore neighborhood. In addition, periodic boundary condition was assumed for simplicity. We denoted the cost of a cell in position $i$ of the lattice as $\Phi_i$. If an intact cell labeled $i$ is surrounded by $l$ damaged cells and $8-l$ intact cells, the expected cost of the intact cell $\Phi_i$ is

$$\Phi_i = \frac{C_{I \leftarrow I}(8-l) + C_{I \leftarrow D}l}{8} \tag{4}$$

Similarly, when the cost of a damaged cell labeled $i$ is surrounded by $l$ damaged cells and $8-l$ intact cells, $\Phi_i$ is

$$\Phi_i = \frac{C_{D \leftarrow I}(8-l) + C_{D \leftarrow D}l}{8} \tag{5}$$

An example of the cost calculation in the lattice-space model is shown in Fig 2. The probability that a cell in position $i$ is eliminated is expressed as

$$\frac{\Phi_i}{\sum_i \Phi_i} \tag{6}$$

Similar to the well-mixed model, we used the waiting time until all cells in the stem cell pool became damaged cells $T_{abs}$ as the criterion for assessing the influence of radiation.

## 2.4 Analyses of the models

For the well-mixed model, $T_{abs}$ was calculated using an approximate solution and 1,000 simulations. Although the results are not shown, 50 simulations were sufficient to calculate the average value. For the lattice space model, 50 simulations were performed with different total cell numbers and compared with the well-mixed model.

## 3. Result

### 3.1 Stem cell competition affects the accumulation of radiation damage under low dose rate irradiation conditions

First, we considered the influence of the dose rate using a well-mixed model, assuming that the initial cell pool is occupied by intact cells (Fig 3a). If all elements in the payoff matrix were equal, e.g. $C_{I \leftarrow I} = C_{I \leftarrow D} = C_{D \leftarrow I} = C_{D \leftarrow D} = 1$, as shown by the black circles, there was no influence of cell–cell interactions because the costs of all cells were equal. This is a neutral situation in which there is no difference between intact and damaged cells. As mentioned above, $\lambda$ corresponded to the dose

(a)

(b)

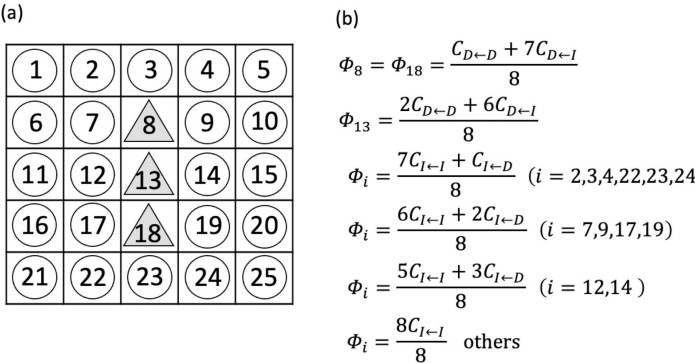

**Fig 2. The abstraction of the lattice-space model.** (a) Cells are arranged in a squared lattice in the lattice-space model. The numbers represent positions in the lattice. We assumed Moore neighborhood and periodic boundary condition. For example, a cell at location 1 interacts with cells at 2, 5, 6, 7, 10, 21, and 22. (b) The cost of each cell in the case of (a). The symbol $\Phi_i$ is the cost of a cell in position $i$.

(a)

(b)

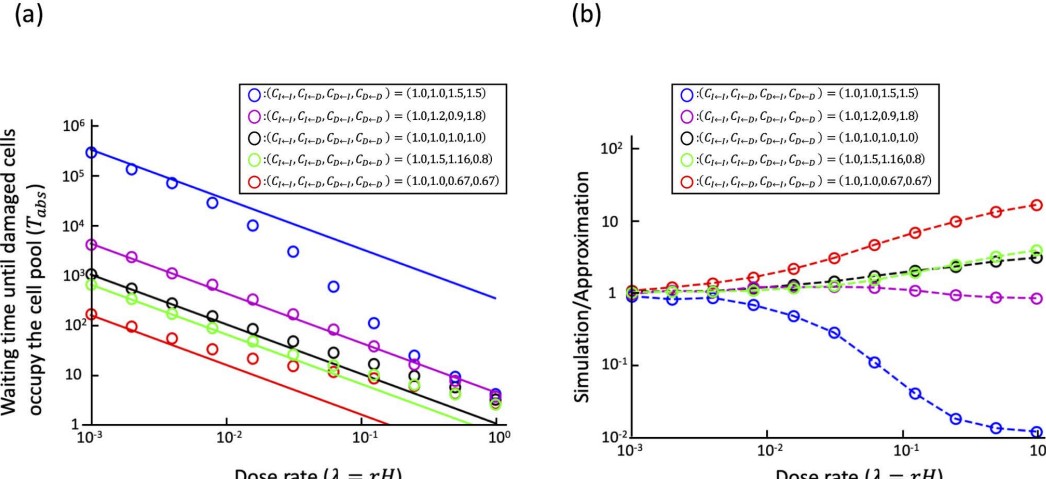

**Fig 3. Waiting time until damaged cells occupy the cell pool, $T_{abs}$, in the well-mixed model.** (a) The horizontal line represents the dose rate $\lambda$. The vertical axis represents $T_{abs}$. Circles indicate the average of 1,000 times simulations. Lines indicate the approximations shown in eq. (7a). The number of cells in the cell pool $N$ is $N = 20$. The parameters of the cost matrix are shown in the figure. Black dots and lines indicate the case in which the costs of all cells are equal, i.e., cell elimination and division occur randomly. We interpret that if $T_{abs}$ is larger than this, the accumulation of damaged cells is suppressed by stem cell competition, and conversely, if $T_{abs}$ is smaller, it is promoted. (b) The ratio of the simulation result to the approximation. Values close to 1 indicate good agreement.

rate. If $\lambda$ was large, $T_{abs}$ was almost the same as in the case of $C_{I\leftarrow I} = C_{I\leftarrow D} = C_{D\leftarrow I} = C_{D\leftarrow D} = 1$. This could be interpreted as the influence of stem cell competition was small under high dose-rate irradiation conditions. Conversely, if $\lambda$ was small and the dose rate was low, the time until the damaged cells occupied the cell pool, was affected by stem cell competition. When $C_{I\leftarrow I} = C_{I\leftarrow D} < C_{D\leftarrow I} = C_{D\leftarrow D}$, as shown by the blue circles, the costs were frequency-independent, and intact cells always had lower costs than damaged cells. In this case, because damaged cells were eliminated more frequently, $T_{abs}$ was greater than when the costs were equal. Conversely, in the case of $C_{I\leftarrow I} = C_{I\leftarrow D} > C_{D\leftarrow I} = C_{D\leftarrow D}$, as shown by the red circles, the costs did not depend on the frequency of each cell, and the costs were lower in the damaged cells. $T_{abs}$ was smaller than that in the case when there was no influence of cell–cell interactions. These results show that the accumulation of damaged cells is suppressed when the dose rate is sufficiently low and damaged cells always have higher costs. Additionally, Fig 3 shows situations in which the cost is frequency-dependent. In such cases, whether intact or damaged cells were more advantageous depended on the situation. Here, we would explain the case of $C_{I\leftarrow I} = 1.0$, $C_{I\leftarrow D} = 1.5$, $C_{D\leftarrow I} = 1.16$, $C_{D\leftarrow D} = 0.8$, as shown by the green circles, as an example. The number of damaged cells induced by irradiation was very small when the dose rate was very low. The influence of the cost of intact cells interacting with each other ($C_{I\leftarrow I}$) and the cost of damaged cells interacting with intact cells ($C_{D\leftarrow I}$) became large. Since $C_{I\leftarrow I} < C_{D\leftarrow I}$, intact cells had a lower cost and were advantageous. The effects of the cost of intact cells interacting with damaged cells ($C_{I\leftarrow D}$) and the cost of interaction between damaged cells ($C_{D\leftarrow D}$) became dominant when the number of damaged cells increased sufficiently. Then, damaged cells had an advantage as $C_{I\leftarrow D} > C_{D\leftarrow D}$. In such cases, cell pool size could be important. An experimental study showed that the result of stem cell competition was affected by the size of the cell pool [32]. Our previous study showed mathematically that the size of the stem cell pool $N$ could affect the results [20]. Next, we investigated the influence of $N$ on $T_{abs}$ by approximation because it was time-consuming to calculate $T_{abs}$ by simulating varying $N$.

### 3.2 Approximation of $T_{abs}$ under very low dose-rate conditions and the influence of the size of stem cell pool

Considering the well-mixed model as an analogy for evolutionary game theory [31], $T_{abs}$ could be approximated under very low dose-rate conditions, i.e., $\lambda$ was very small, as follows:

$$T_{abs} = \frac{1}{N\lambda\pi_1},$$

(7a)

where,

$$\pi_1 = \frac{1}{1 + \sum_{j=1}^{N-1} \prod_{k=1}^{j} \frac{G_k}{F_k}}.$$

(7b)

The variable $\pi_1$ was the probability that starting from one damaged cell, the damaged cell would occupy the cell pool without becoming extinct. The derivation of $\pi_1$ is shown in Appendix B. The size of the cell pool $N$ and the indicators of dose rate $\lambda$ were given regardless of the presence or absence of cell–cell interactions. It was $\pi_1$ that expressed the effect of cell–cell interactions. This approximation was consistent with the simulation results when $\lambda$ was small (Fig 3a and 3b). The dependence of the approximated $T_{abs}$ on $N$ was qualitatively classified into four patterns (Fig 4). Fig 4A shows that $T_{abs}$ was constant regardless of $N$. This pattern appeared when the neutral situations where $C_{I\leftarrow I} = C_{I\leftarrow D} = C_{D\leftarrow I} = C_{D\leftarrow D}$. This is because when $C_{I\leftarrow I} = C_{I\leftarrow D} = C_{D\leftarrow I} = C_{D\leftarrow D}$, $\pi_1 = \frac{1}{N}$, and $T_{abs}$ is independent of $N$. Therefore, we normalized $T_{abs}$ to 1 when $C_{I\leftarrow I} = C_{I\leftarrow D} = C_{D\leftarrow I} = C_{D\leftarrow D} = 1$. Thus, stem cell competition suppressed the accumulation of damaged

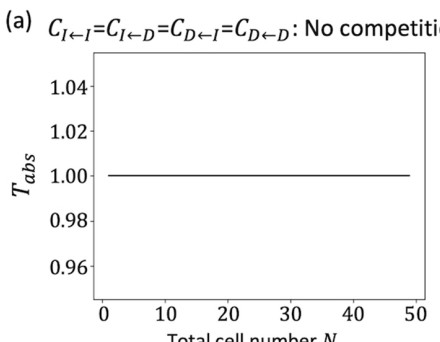
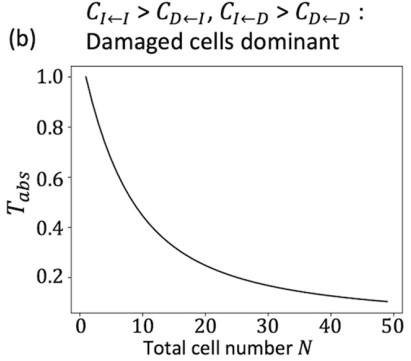
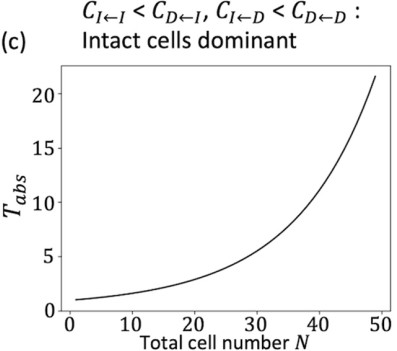

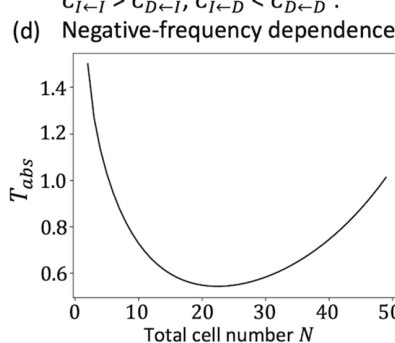
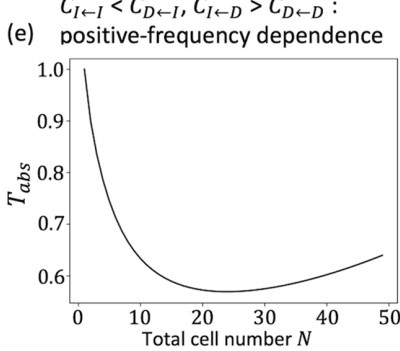

**Fig 4. Dependence $T_{abs}$ on the total number of cells.** The horizontal line represents the number of cells in the cell pool $N$. The value of $T_{abs}$ is normalized as 1 for $C_{I\leftarrow I} = C_{I\leftarrow D} = C_{D\leftarrow I} = C_{D\leftarrow D} = 1.0$. The vertical axis represents the $T_{abs}$ calculated under a very low dose rate condition in eq. (7a). (a) $C_{I\leftarrow I} = C_{I\leftarrow D} = C_{D\leftarrow I} = C_{D\leftarrow D} = 1.0$, (b) $C_{I\leftarrow I} = C_{I\leftarrow D} = 1.0$, $C_{D\leftarrow I} = C_{D\leftarrow D} = 0.8$, (c), $C_{I\leftarrow I} = C_{I\leftarrow D} = 1.0$, $C_{D\leftarrow I} = C_{D\leftarrow D} = 1.25$, (d) $C_{I\leftarrow I} = 1.0$, $C_{I\leftarrow D} = 0.25$, $C_{D\leftarrow I} = 0.5$, $C_{D\leftarrow D} = 0.75$ and (e) $C_{I\leftarrow I} = 1.0$, $C_{I\leftarrow D} = 1.5$, $C_{D\leftarrow I} = 1.2$, $C_{D\leftarrow D} = 0.5$.

cells when $T_{abs} > 1$ and promoted accumulation when $T_{abs} < 1$. Fig 4B shows that $T_{abs}$ decreased monotonically depending on $N$; this included the cases where the costs were frequency independent and the intact cell had a greater cost, $C_{I \leftarrow I} = C_{I \leftarrow D} > C_{D \leftarrow I} = C_{D \leftarrow D}$. Conversely, when the cost was frequency independent and the damaged cell had a greater cost, $C_{I \leftarrow I} = C_{I \leftarrow D} < C_{D \leftarrow I} = C_{D \leftarrow D}$, $T_{abs}$ increased monotonically depending on $N$ (Fig 4C). When $N$ is small, cells with lower cost were likely eliminated by chance, and cells with higher cost could occupy the cell pool. Conversely, when $N$ is large, such accidental effects are small, and the more advantageous effects tend to expand in the cell pool. Therefore, when damaged cells had a higher cost, $T_{abs}$ increased with increasing $N$. On the other hand, $T_{abs}$ decreased with increasing $N$ when intact cells were disadvantageous. Fig 4D and 4E show a downward convex shape depending on $N$. $T_{abs}$ decreased with increasing $N$, but when $N$ reached a certain degree, it increased.

Fig 5 illustrates the shape of $T_{abs}$ when $C_{I \leftarrow I} = C_{I \leftarrow D} = 1$. The cases when $C_{I \leftarrow D}$ was changed are listed in S1 Fig. If $C_{I \leftarrow I} < C_{D \leftarrow I}$, the cost of the intact cell interacting with an intact cell is smaller than that of the damaged cell interacting with an intact cell. Intact cells are more advantageous when interacting with intact cells. Conversely, if $C_{I \leftarrow I} > C_{D \leftarrow I}$, the damaged cells are more advantageous when interacting with the intact cells. The vertical line in the diagram shows $C_{I \leftarrow I} = C_{D \leftarrow I}$. Assuming an intact cell or a damaged cell interacting with an intact cell, the intact cell have an advantage on the right side and the damaged cell have an advantage on the left side. Similarly, the horizontal line in the diagram shows $C_{I \leftarrow D} = C_{D \leftarrow D}$. When an intact cell or a damaged cell interact with a damaged cell, the intact cells have an advantage

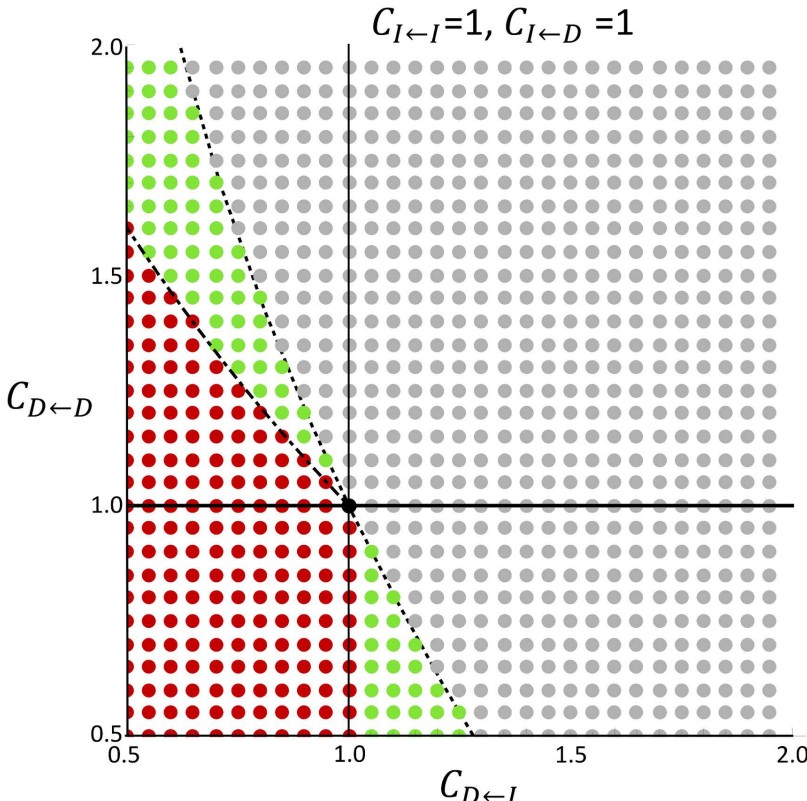

**Fig 5. Summary of the shape of $T_{abs}$.** Fixing $C_{I \leftarrow I} = C_{I \leftarrow D} = 1$. Gray, red, and green dots represent monotonic increase, monotonic decrease, and convex downward, respectively. Black dot represents the case that $T_{abs}$ did not change depending on $N$. Vertical and horizontal lines in the diagram represent $C_{D \leftarrow I} = C_{I \leftarrow I}$ and $C_{D \leftarrow D} = C_{I \leftarrow D}$, respectively. The dotted line represents the parameters $T_{abs}$ decreases when $N$ changes from $N = 2$ to $N = 3$. The dotted-dashed line represents the parameters $T_{abs}$ increases when $N$ is very large, in which $N = 10^5$ hear.

above this line ($C_{I \leftarrow D} < C_{D \leftarrow D}$), and the damaged cells have an advantage below this line ($C_{I \leftarrow D} > C_{D \leftarrow D}$). These lines divide the diagram into four regions. In the upper-right region, intact cells are always at an advantage, and $T_{abs}$ increased monotonically with $N$. Conversely, in the lower-left region, damaged cells are always at an advantage, and $T_{abs}$ decreased monotonically. In terms of game theory, an intact cell is the dominant strategy in the upper right. We will refer to this condition as "intact cell dominant". On the other hand, a damaged cell is the dominant strategy in the lower left. This condition will be referred to as "damaged cell dominant". A typical quner's dilemma situation would correspond to these conditions. In the lower right region, $C_{I \leftarrow I} < C_{D \leftarrow I}$ and $C_{I \leftarrow D} > C_{D \leftarrow D}$, the damaged cells have higher costs than the intact cell when interacting with intact cells, and intact cells have higher costs when interacting with damaged cells. It can be interpreted that the cost of interacting with one type of cell is higher than that of the cells interacting with a different type of cell. We will refer to this condition as "positive-frequency dependence". This corresponds to a coordination game in game theory. The shape of $T_{abs}$ was convex downward if $C_{D \leftarrow I}$ close to $C_{I \leftarrow I}$, and became a monotonically increasing pattern when $C_{D \leftarrow I}$ became larger. Initially, very few damaged cells are observed, and both intact and damaged cells tend to interact with intact cells. Then, $C_{I \leftarrow I}$ and $C_{D \leftarrow I}$ can be compared. If $C_{I \leftarrow I} < C_{D \leftarrow I}$, damaged cells have more cost in this situation. They are more likely to be eliminated and less likely to increase. Conversely, the number of damaged cells increase more easily when $C_{I \leftarrow D} > C_{D \leftarrow D}$. This indicates that it is difficult to increase the number of damaged cells when the number of damaged cells is small. However, when the number of damaged cells increased, it is easy to increase the number of damaged cells. If $N$ is small, the number of damaged cells increased stochastically to that extent; therefore, $T_{abs}$ decreased until $N$ reached a certain level. When $N$ is large, it took time for the number of damaged cells to increase to that extent; therefore $T_{abs}$ became larger depending on $N$. In the upper-left region, $C_{I \leftarrow I} > C_{D \leftarrow I}$ and $C_{I \leftarrow D} < C_{D \leftarrow D}$, the intact cells have a higher cost than the damaged cell when interacting with the intact cells, and the damaged cells have a higher cost when interacting with the damaged cells. This can be interpreted as the cost of interacting with one type of cell being greater for cells of the same type and that interacting with cells of different types is more advantageous. We will refer to this condition as "negative-frequency dependence". This condition includes the hawk-dove game. If $C_{D \leftarrow I}$ is sufficiently small, $T_{abs}$ was monotonically decreased depending on $N$. If $C_{D \leftarrow I}$ close to $C_{I \leftarrow I}$, the shape of $T_{abs}$ was monotonically increased. The shape of $T_{abs}$ was convex downward when $C_{D \leftarrow I}$ was intermediate. As mentioned above, $C_{I \leftarrow I}$ and $C_{D \leftarrow I}$ is compared in the initial stage. Because the cost of intact cells is higher in this situation, the number of damaged cells tend to increase. As the number of damaged cells increase, a comparison could be made between $C_{I \leftarrow D}$ and $C_{D \leftarrow D}$. Then, the increase of damaged cells is less frequent since $C_{I \leftarrow D} < C_{D \leftarrow D}$. When $N$ is small, the former effect was dominant until the number of damaged cells reached a certain level, and the latter effect stochastically eliminated the advantageous intact cells. Therefore, $T_{abs}$ decreased depending on $N$ up to a certain value. If $N$ is sufficiently large, the stochastic effect is small. As a result, the larger $N$ was, the larger $T_{abs}$ became. To distinguish between monotonically decreasing and downward convexity, $N$ must be large enough. However, it is difficult to verify this numerically. Therefore, we calculated the boundary that decrease when $N$ is small and the boundary that increase when $N$ is large. The boundaries shown in Fig 5 and the S1 Fig were calculated in Appendix C and agree well with the numerically calculated boundaries.

### 3.3 Spatial structure can qualitatively change the dependence of $T_{abs}$ on the size of the cell pool

Since the analytical calculation of $\pi_1$ is difficult when assuming a spatial structure, we calculated $\pi_1$ by simulation. We set a single damaged cell in a cell pool and repeated the cell elimination and division events until all the cells became damaged cells or intact cells. Then, we obtained the value of $\pi_1$ as the frequency of the case in which all the cells became damaged cells. We compared the well-mixed and lattice space models with the parameter sets of the above five patterns (Fig 6). In neutral case, $C_{I \leftarrow I} = C_{I \leftarrow D} = C_{D \leftarrow I} = C_{D \leftarrow D} = 1$, there was no difference between the intact cell and the damaged cell, and there was no difference between the well-mixed model and the lattice space model (Fig 6A). In damaged cells dominant, $C_{I \leftarrow I} > C_{D \leftarrow I}$, $C_{I \leftarrow D} > C_{D \leftarrow D}$, $T_{abs}$ of the lattice space model also decreased monotonically with increasing $N$ (Fig 6B). Conversely, in intact cells dominant, $C_{I \leftarrow I} < C_{D \leftarrow I}$, $C_{I \leftarrow D} < C_{D \leftarrow D}$, $T_{abs}$ of the lattice space model

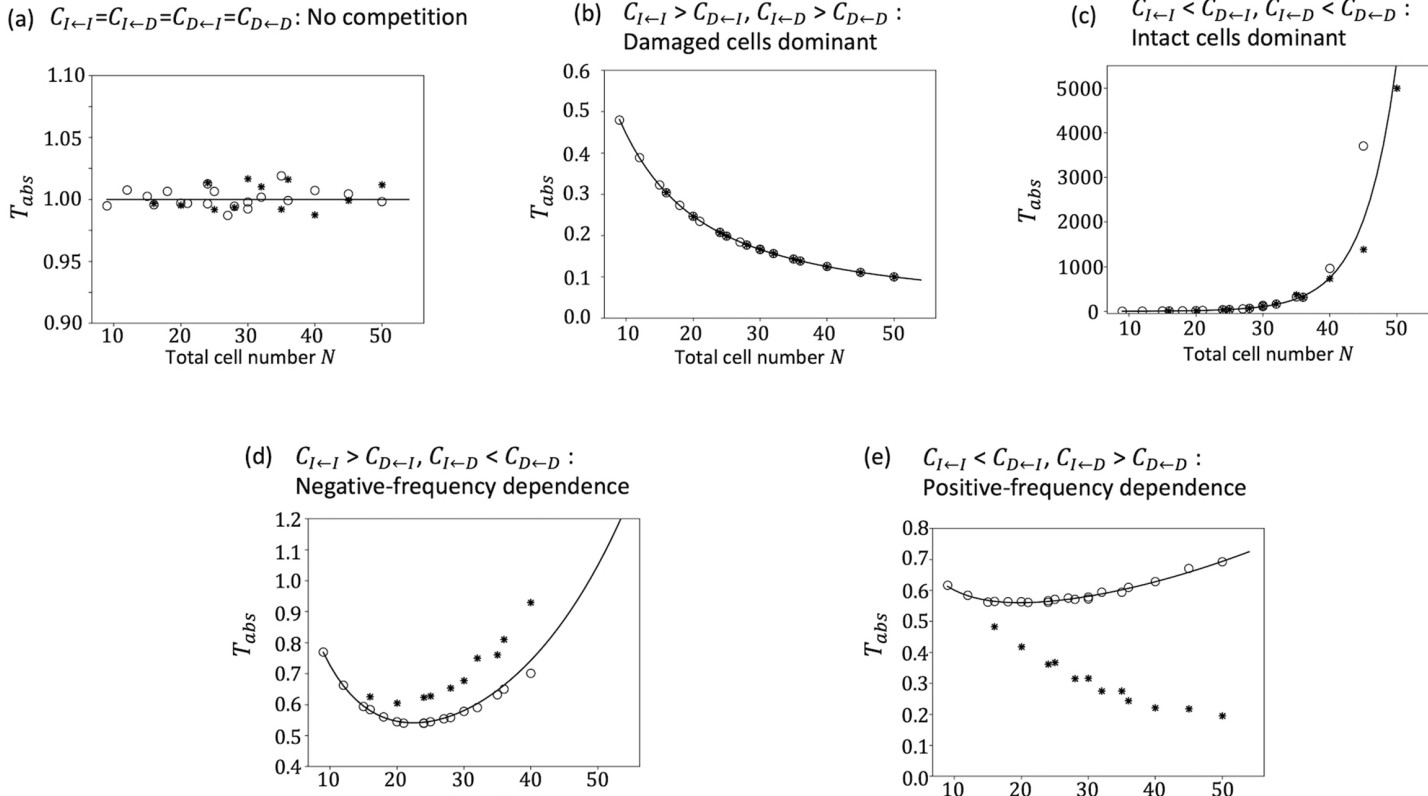

**Fig 6. Comparison of the well-mixed model and the lattice space model.** White circles and asterisks are the average of 50 simulations of the well-mixed and the lattice space models, respectively. The black line is the approximated $T_{abs}$ under the very low dose rate condition shown in eq. (7a). (a) $C_{I\leftarrow I} = C_{I\leftarrow D} = C_{D\leftarrow I} = C_{D\leftarrow D} = 1.0$, (b) $C_{I\leftarrow I} = C_{I\leftarrow D} = 1.0$, $C_{D\leftarrow I} = C_{D\leftarrow D} = 0.8$, (c), $C_{I\leftarrow I} = C_{I\leftarrow D} = 1.0$, $C_{D\leftarrow I} = C_{D\leftarrow D} = 1.25$, (d) $C_{I\leftarrow I} = 1.0$, $C_{I\leftarrow D} = 0.25$, $C_{D\leftarrow I} = 0.5$, $C_{D\leftarrow D} = 0.75$ and (e). $C_{I\leftarrow I} = 1.0$, $C_{I\leftarrow D} = 1.75$, $C_{D\leftarrow I} = 1.25$, $C_{D\leftarrow D} = 0.5$..

increased monotonically with increasing $N$ similar to that in case of the well-mixed model (Fig 6C). These two results were qualitatively consistent with the well-mixed model; quantitatively, there was little difference between them. When $C_{I\leftarrow I} = 1.0, C_{I\leftarrow D} = 0.25, C_{D\leftarrow I} = 0.5, C_{D\leftarrow D} = 0.75$, the shape of $T_{abs}$ was convex downward depending on $N$ as the well-mixed model, although the value was larger than that of the well-mixed model (Fig 6D). This implies that damaged cells occupied the cell pool more quickly in the lattice space model than in the well-mixed model. This parameter set is negative-frequency dependent ($C_{I\leftarrow I} > C_{D\leftarrow I}$, $C_{I\leftarrow D} < C_{D\leftarrow D}$) and represents a scenario where a cell of one type is more advantageous when interacting with a cell of another type. The spatial structure increased the opportunities for interactions between cells of the same type. Therefore, if $C_{D\leftarrow D}$ is sufficiently smaller than $C_{I\leftarrow I}$, the damaged cells will be harder to eliminate and will spread more quickly throughout the entire area than in the case where there is no spatial structure. When $C_{I\leftarrow I} = 1.0, C_{I\leftarrow D} = 1.75, C_{D\leftarrow I} = 1.25, C_{D\leftarrow D} = 0.5$, the results were qualitatively different between the well-mixed model and the lattice space model (Fig 6E). $T_{abs}$ in the well-mixed model had a convex downward shape depending on $N$; however, $T_{abs}$ in the lattice space model decreased monotonically. This parameter set is an example of positive-frequency dependent, $C_{I\leftarrow I} < C_{D\leftarrow I}$, $C_{I\leftarrow D} > C_{D\leftarrow D}$. That is, the cells that interact with different cell types have a greater cost. In the absence of a spatial structure and when there are fewer damaged cells, the influence of interactions between damaged cells decreases as $N$ increases. In contrast, when there is a spatial structure, the damaged cells are adjacent to each other after division; therefore, even if there are only a few damaged cells, the interaction between the damaged cells can

have considerable impact. In this case, the cost of damaged cells is lower than that in the case with no spatial structure, and they are less likely to be eliminated. Even if $N$ increases, the damaged cells will still be adjacent to each other; therefore, the situation favoring damaged cells will continue, and $T_{abs}$ will become smaller.

## 4. Discussion

We have shown that the accumulation of damaged cells was suppressed by stem cell competition under low dose-rate irradiation conditions when the damaged cell was disadvantaged [20]. In this study, we developed a mathematical model based on the Moran model that explicitly handles the case in which the cost, which could be interpreted as the inverse of the fitness, depended on the frequency of other cells. In order to discuss the effects of radiation, this paper assumes stem cells, but the mathematical model can also be applicable to cell competition more generally. Some of the results of the proposed model were consistent with those of our previous model. For example, the influence of cell competition was strong under low dose-rate condition; however, the influence was small when the dose rate was high (Fig 3). Furthermore, we could approximate the average waiting time for the damaged cells to occupy the cell pool, $T_{abs}$, when the dose rate was very low. The large $T_{abs}$ means that it would take a long time for damage accumulation. In that case, the carcinogenic effects of radiation would be negligible. An elemental dose, which is the lowest dose delivered by a single track of radiation to the nucleus of a cell, of γ-rays is approximately 1 mGy for the typical mammalian cell nucleus [33]. Assuming an 8-hour workday and 250 working days per year (i.e., 5 days per week for 50 weeks), the radiation exposure during work is 0.01 mGy/h (=20 mGy ÷ [250 days x 8 hours]) to reach 20 mGy per year, which is the average effective dose limit for workers. Under these conditions, it would take 100 hours of exposure to reach a cumulative dose of 1 mGy, which is equivalent to approximately 12.5 working days at 8 hours per day. When the cell cycle length was 24 h, an average of one hit occurred during 12.5 divisions. In our model, the dose rate $\lambda$ represents an average of $\lambda$ hits per cell per cell cycle length. Assuming conservatively that elemental dose always causes damage, $H = 1$, one hit per cell after 12.5 divisions means $12.5\lambda = 1 \rightarrow \lambda = 0.08$. Whether the approximation is valid at this dose rate depends on the parameters (e.g., Fig 3B), but even at dose rates where the approximate is not valid, it may be possible to use it as an upper or lower bound on the waiting time for the damaged cells to occupy the cell pool. In the following, we discuss the case of a sufficiently low dose rate, for which the approximate solution is justified. If the cost is frequency-independent, it would be clear whether intact or damaged cells are advantageous. However, assuming a frequency-dependent cost, the cost of a cell depends on both its type and the types of cells with which it interacts. Therefore, whether an intact or damaged cell has an advantage depends largely on the situation. Some mutant cells can eliminate the wild-type cells if the number of mutant cells is large enough [11]. The difference between frequency-independent and frequency-dependent costs was clear when we examined how the accumulation of radiation damage changed according to the total number of cells (Fig 4). When we determined whether damaged or intact cells were advantageous, $T_{abs}$ changed monotonically with the total cell number (Fig 4B and 4C). Therefore, the total number of cells does not have any qualitative influence on whether stem cell competition suppresses or promotes the accumulation of damaged cells. However, when we assumed a frequency-dependent cost, it was observed that the influence of cell competition could drastically change depending on the total number of cells (Fig 4D and 4E). Even if the cost matrix is the same, cell competition may promote damage accumulation when the total number of cells is small and suppress it when the total number of cells is large. Although it has been experimentally confirmed that the cell pool size affects the outcome of cell competition [32], our results imply that differences in the cell pool size may qualitatively alter the impact of stem cell competition on the accumulation of radiation damage. The number of stem cells varies substantially among tissues. For example, the intestine contains a large number of crypts, and the number of stem cells per crypt is estimated to be approximately 14 in the small intestine [34,35] and approximately 6 in the colon [34,36]. In contrast, the number of hematopoietic stem cells is estimated to be $1.35 \times 10^8$ [37]. Even if the cell properties are the same, the effects of stem cell competition may differ among tissues.

The spatial structure is known to influence the results of the game in the field of evolutionary biology [28,29]. Cells in the living body also make some kinds of spatial structure, and cell competition can occur by short-range cell interaction. When the cost was frequency-independent, the spatial structure did not change the results (Fig 6A–6C). However, the cost was frequency-dependent, and the results could have been influenced by the spatial structure (Fig 6D and 6E). In particular, the results were qualitatively different when the cost of interacting with a same cell type was lower for the different cell type (Fig 6E). In contrast to the well-mixed model in which the cost is determined by the frequency of cells in the entire population, the cost of the lattice-space model is determined only through interactions among adjacent cells. Therefore, although damaged cells are likely to be eliminated by interaction with intact cells in the well-mixed model, the effect of the interaction with intact cells can be suppressed by creating clusters among damaged cells in the lattice-space model. Recently, it has become possible to quantitatively estimate the effects of cell competition from experimental data and mathematical models [17–19]. For example, in the co-cultures of non-small cell lung cancer cells that are sensitive and resistant to anticancer drugs, it has been shown that the presence of cancer-associated fibroblasts alters parameters related to cell competition. [17]. If it were possible to observe a situation in which the number of cells remains constant, it might be possible to estimate the parameters of the cost matrix assumed in this model. On the other hand, these models did not assume a spatial structure, considering that the spatial structure may not be necessary when cells actively move around under culture conditions. However, when discussing *in vivo* conditions with experimental data, it is necessary to consider the spatial structure of the tissue of interest, especially when the cost and fitness of the cells are frequency dependent.

In the cell competition experiments, there was no difference between the growth of irradiated and non-irradiated cells when they were not mixed; however, when they were mixed, the growth of irradiated cells was suppressed [13,14]. From the results obtained when the cells were not mixed, it can be interpreted as $C_{I \leftarrow I} = C_{D \leftarrow D}$ in our model. Since the growth of irradiated cells was suppressed when mixed with non-irradiated cells, it is interpreted that $C_{I \leftarrow D} < C_{D \leftarrow I}$. Given $C_{I \leftarrow I} = C_{D \leftarrow D}$ and $C_{I \leftarrow D} < C_{D \leftarrow I}$, there are three possible patterns for the magnitude of the parameters: $C_{D \leftarrow I} > C_{I \leftarrow I} = C_{D \leftarrow D} > C_{I \leftarrow D}$, $C_{D \leftarrow I} > C_{I \leftarrow D} > C_{I \leftarrow I} = C_{D \leftarrow D}$, and $C_{I \leftarrow I} = C_{D \leftarrow D} > C_{D \leftarrow I} > C_{I \leftarrow D}$. If $C_{D \leftarrow I} > C_{I \leftarrow I} = C_{D \leftarrow D} > C_{I \leftarrow D}$ (shown in the upper-right region of Figs 5 and S1), intact cells are always advantageous. In this case, the accumulation of damaged cells was suppressed by cell competition, and the presence or absence of a spatial structure did not have a significant influence on the results. The parameter set $C_{D \leftarrow I} > C_{I \leftarrow D} > C_{I \leftarrow I} = C_{D \leftarrow D}$ (shown in the lower-right region of Figs 5 and S1) indicates that the interaction with different types of cells has a lower cost. Depending on these parameters, whether competition suppresses or promotes the accumulation of damaged cells can be reversed by changing the size of the cell pool. A spatial structure causes a quantitative difference in the results, but not a qualitative difference. If $C_{I \leftarrow I} = C_{D \leftarrow D} > C_{D \leftarrow I} > C_{I \leftarrow D}$ (shown in the upper-left region of Figs 5 and S1), interaction with the same types of cells have a lower cost. In this case, as shown in Fig 6E, the results may be qualitatively different, depending on whether there is a spatial structure. Although it is important to estimate the parameters related to cell competition, one must be very careful about how the spatial structure affects the results when discussing the *in vivo* effects of these parameters.

Presented model is very simple and there is room for improvement. For example, it is unclear how damaged and intact cells coexist depending on the dose rate. In addition, radiation only irreversibly damages cells with a certain probability. This assumption does not explicitly consider repair mechanisms. Furthermore, the cost matrix parameters and the cell pool size are not affected by different dose rates. They may change between high-dose and low-dose-rate exposures. For example, if radiation exposure makes damaged cells more susceptible to elimination, then $C_{I \leftarrow D}$ would be an decreasing function of $\lambda$, whereas $C_{D \leftarrow I}$ and $C_{D \leftarrow D}$ would be increasing functions of $\lambda$ ($\frac{dC_{I \leftarrow D}}{d\lambda} \leq 0$, $\frac{dC_{D \leftarrow I}}{d\lambda} \geq 0$, and $\frac{dC_{D \leftarrow D}}{d\lambda} \geq 0$). Conversely, if damaged cells become less likely to be eliminated, $C_{I \leftarrow D}$ would be an increasing function of $\lambda$, while $C_{D \leftarrow I}$ and $C_{D \leftarrow D}$ could be represented as decreasing functions of $\lambda$ ($\frac{dC_{I \leftarrow D}}{d\lambda} \geq 0$, $\frac{dC_{D \leftarrow I}}{d\lambda} \leq 0$, and $\frac{dC_{D \leftarrow D}}{d\lambda} \leq 0$). Mathematically, various assumptions can be made; therefore, the specific functional forms may be examined experimentally, for instance by

conducting co-culture experiments in which cells are irradiated under different dose rates. Radiation-induced carcinogenesis is a complex process involving DNA repair, cell-cell interaction, and alterations in the tissue microenvironment, which have recently attracted attention [38]. The presented model explicitly captures how cell-cell interactions affects the costs associated with cell proliferation. To explain why low-dose-rate exposure does not necessarily increase cancer risk, future models will need to incorporate how these other systems respond differently across dose rates.

## 5. Conclusions

In conclusion, we aimed to develop a mathematical model that assumes frequency-dependent costs, which will be negatively correlated with cell proliferation, to account for the influence of stem cell competition on radiation damage accumulation. In the absence of a spatial structure, cell competition suppressed damage accumulation as the total number of cells increased when intact cells were advantageous and accelerated it when damaged cells were advantageous. However, when the advantageous cells varied depending on the situation, whether the competition suppressed or promoted damage accumulation could be reversed, depending on the total number of cells. In recent years, the effects of cell-cell interactions have begun to be measured quantitatively. Considering the number of cells in the population will likely deepen our understanding of cell competition.

Furthermore, when we assumed a spatial structure, the effects of competition were qualitatively different in some cases compared with cases without a spatial structure. Even if damage accumulation is assumed to be suppressed under culture conditions in which the spatial structure can be ignored, damage accumulation may be promoted if the spatial structure exists *in vivo*. When considering the effects of stem cell competition on radiation damage in detail, it is important to consider not only the properties of cell interactions but also the size of the cell pool and its spatial structure *in vivo*. It will be difficult to measure spatial structures *in vivo*, and simulations using mathematical models are useful method. Although the mathematical model in this paper differs from the actual structure *in vivo*, we were able to discuss in what cases the spatial structure has an effect. This type of analysis will be useful when investigating structures that are closer to those *in vivo*.

Radiation carcinogenesis involves multiple biological processes. In this study, we focused specifically on stem cell competition, a form of cell–cell interaction, and the model allowed us to discuss the conditions under which such competition suppresses the accumulation of damage. Future extensions of the model that incorporate additional biological processes will provide a more comprehensive framework for evaluating cancer risk from low-dose-rate radiation exposure.

## Supporting information

**S1 Fig.  Summary of the shape of $T_{abs}$ varying $C_{I \leftarrow D}$, $C_{D \leftarrow I}$ and, $C_{D \leftarrow D}$.** The parameter $C_{I \leftarrow I}$ is fixed as $C_{I \leftarrow I} = 1$ without loss of generality. The gray, red, and green dots represent monotonic increase, monotonic decrease, and convex downward, respectively. The black dot represents the case that $T_{abs}$ did not change depending on $N$. The vertical and horizontal lines in the diagram represent $C_{D \leftarrow I} = C_{I \leftarrow I}$ and $C_{D \leftarrow D} = C_{I \leftarrow D}$, respectively. The dotted line represents the decrease in parameter $T_{abs}$ when $N$ changes from $N = 2$ to $N = 3$. The dotted-dashed line represents the increase in parameter $T_{abs}$ when $N$ is very large (here, $N = 10^5$).
(TIF)

**S1 File.  Appendices A, B, and C.**
(DOCX)

## Acknowledgments

We thank Dr. KI for technical discussions on the mathematical model.

## Author contributions

**Conceptualization:** Kouki Uchinomiya, Masanori Tomita.

**Data curation:** Kouki Uchinomiya, Masanori Tomita.

**Formal analysis:** Kouki Uchinomiya.

**Funding acquisition:** Kouki Uchinomiya, Masanori Tomita.

**Investigation:** Kouki Uchinomiya, Masanori Tomita.

**Methodology:** Kouki Uchinomiya, Masanori Tomita.

**Project administration:** Kouki Uchinomiya, Masanori Tomita.

**Resources:** Kouki Uchinomiya, Masanori Tomita.

**Software:** Kouki Uchinomiya.

**Supervision:** Masanori Tomita.

**Validation:** Kouki Uchinomiya, Masanori Tomita.

**Visualization:** Kouki Uchinomiya, Masanori Tomita.

**Writing – original draft:** Kouki Uchinomiya, Masanori Tomita.

**Writing – review & editing:** Kouki Uchinomiya, Masanori Tomita.

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
