## [Decision Letter · Decision Letter 0]

11 Jun 2025

Dear Dr. Uchinomiya,

We look forward to receiving your revised manuscript.

Kind regards,

Attila Csikász-Nagy

Academic Editor

PLOS ONE

2. Please expand the acronym “JSPS” (as indicated in your financial disclosure) so that it states the name of your funders in full.

Reviewers' comments:

Reviewer's Responses to Questions

**Comments to the Author**

1. Is the manuscript technically sound, and do the data support the conclusions?

Reviewer #1: Yes

Reviewer #2: Partly

Reviewer #3: Yes

2. Has the statistical analysis been performed appropriately and rigorously?

Reviewer #1: N/A

Reviewer #2: N/A

Reviewer #3: Yes

3. Have the authors made all data underlying the findings in their manuscript fully available?

Reviewer #1: No

Reviewer #2: Yes

Reviewer #3: Yes

4. Is the manuscript presented in an intelligible fashion and written in standard English?

Reviewer #1: Yes

Reviewer #2: Yes

Reviewer #3: Yes

Reviewer #1: Uchinomiya and Tomita present a mathematical model of the effect of competition between “intact” and “damaged” stem cells on the accumulation of damaged cells within the population. Interestingly, although the text is focused on low-dose radiation exposure, the model itself is not specific to radiation. Overall, I enjoyed reading the manuscript, but offer some comments and suggestions below.

Major comments

1. According to the introduction, the goal of this study is to investigate the influence of stem cell competition on accumulation of damage from low dose background radiation. However, the discussion and conclusion do not clearly or strongly tie the results back to this goal. That is, there is no discussion of what implications these results have for protective measures that may be taken against low dose rate exposure.

2. Lines 380-389. The discussion presented in these lines does not make sense. Please completely rewrite this section. Firstly, as far as this reviewer knows and can find, there exists no such definition of an “elemental dose” in this context. In addition, reference 8, which is given for this definition, does not discuss such a concept. What is the correct reference? Second, the calculation that it would take 18 days to reach 1mGy under the assumption of an exposure limit of 20mGy per year isn’t valid. The time it takes to accumulate any exposure would depend solely on the dose rate being experienced, not the upper limit of dose exposure. Third, it is not clear what approximation the authors are referring to in line 387.

3. During model development, several explicit or implicit assumptions were made that need to be discussed either in the methods or in the discussion section. For example, it is assumed that damage is irreversible. However, it is well known that radiation induced DNA damage is repairable and could even occur on the time scales considered in the article.

Minor comments

1. Lines 217-227. The text refers to data presented in Figure 3. There is a mismatch between the color of the data in the figure that corresponds to the values (green) and the color used in the text (pink).

2.Please provide information regarding the initial makeup of the population. Do the simulations begin with one damaged cell, or only intact cells?

3. The definition of Tabs isn’t clear. Is it when there is only damaged stem cells present?

4. Can the lattice-based model only support a certain number of cells, or does the lattice expand indefinitely?

5. How did the authors decide on performing 50 simulations per condition? Currently the first mention that 50 simulations were performed is in the caption of Figure 3. Please also mention this in the methods section. Further, please adapt the figures (where appropriate) to include the standard deviation as error bars.

6. Please include a brief explanation of how to the C values can be interpreted. For example, does C_(I←D)<1 indicate that damaged cells lower or increase the cost of intact cells?

7. The manuscript mentions “low dose-rate” conditions as being when λ is “very small”. In Figure 3, λ ranges from 10^-3 to 1. Are all of these values considered to be low dose rate?

8. According to lines 252-254 “…Tabs could be normalized to 1” in Figure 4 (a). Has Tabs been normalized to one in the rest of Figure 4, or has it been normalized to the value of Tabs from Figure 4(a)? It is not clear.

9. Lines 377-378 states “For example, the influence of cell competition was significant under low dose-rate condition..”. Please clarify if this refers to a statistically significant difference or not.

10. Lines 93, 134, 190. Please replace the word abstract with abstraction.

11. Line 109-110 is not clear. Please rephrase.

12. Lines 163-164 state “The well-mixed model is a Markov process, and the absorbing state is only when damaged cells occupy the cell pool.” Do the authors have a proof for this statement? If so, please include it in the appendix or supplemental information.

13. Line 175-176 “Then, one cell is randomly chosen from the removed cell and surrounding cells to divide and fill the removed space.” So the removed cell proliferates? Please clarify.

14. I encourage the authors to be consistent with the tenses used throughout the manuscript. In some places they have used past tense, and in some they use present tense.

15. Lines 255-256 and 258-259. I believe the authors have switched the descriptions. In Figure 4(b), Tabs monotonically decreases with N, and in Figure 4(c) it monotonically increases with N.

16. In line 292, there is a typo. I believe “closed” should be “close”.

17. In line 302, it should be “therefore”.

18. The description of figure 5, and how the dotted and dash dotted lines were chosen and what they mean is unclear. Please rephrase the explanation.

Reviewer #2: Title: A Mathematical Model assuming Frequency-dependent Cost for Analyzing the Influence of Cell Competition on Radiation Effects

Summary: authors use standard Moran process with state-switching model to determine the influence of radiation on stem cell damaged cell’s expanding over intact stem cells. Then the authors extend the model to a standard spatial Moran process with 8 nearest neighbors.

Overall the research question is important, but the methods are standard and the connection to existing theoretical literature is weak, including any discussion of types of payoff matrices (e.g. prisoner’s dilemma; hawk-dove; etc) rather than relying on hard-coded payoff values. There are some theoretical analysis of the rate of mutant expansion that might be helpful here, too. For a good example, check Heyde et. al. Cell, 2021.

The most interesting part of the manuscript is the connection between frequency-dependent competition and radiation dose. However the dose parameter, \lambda, is not contained in the equations of the competition model in equation 3, making it difficult to determine its influence. Secondly, I assume that \lambda will have an effect on stem cell pool size, N, and that radiation may influence the competition parameters in the matrix too.

One of the central claims of the paper, that “When the dose rate is very low, some stem cells may be damaged and mixed with intact stem cells in the stem cell pool” isn’t sufficiently explored in the model. For example, an effect of radiation dose is linear, and thus there is no qualitative regime change between high and low doses of radiation.

Many claim in the introduction or conclusion section are true claims, but do not have associated references; for example:

1. Line 36: “Ionizing radiation is a well-known external factor that increases the risk of cancer. “

2. Line 59: “The hypothesis that stem cell competition suppresses the effects of radiation has been supported by experiments.”

3. Line 462: “In recent years, the effects of cell-cell interactions have begun to be measured quantitatively.”

Reviewer #3: I have read the article “A Mathematical Model assuming Frequency-dependent Cost for Analyzing the Influence of Cell Competition on Radiation Effects.” In this manuscript, the authors analyze the effect of cell-cell competition on the accumulation of cells damaged by radiation. They constructed mathematical model of radiation damage which used a Moran process to model growth and death of cells, a Markov model for transition probabilities between induced and damaged cells, and an evolutionary game of interaction between cells. They analyzed this under a well-mixed and spatially explicit model. They showed that frequency-dependent competition between cells can slow down the rate of damaged cell takeover, at least under low-dose radiation. Overall, this is a good paper with an appropriate model that shows interesting results. I cannot find serious flaws with the methodology or analysis. I list my comments regarding the manuscript below.

My first comment is that elements of the manuscript could be made more clearly. Firstly, the authors mention cell fitness. It would be good to define fitness in the context of cells. I have an understanding of what they mean but a formal definition would help. They also mention stem cell competition but it seems like this model could be broadened to any kind of cell. I would either state the reasons for focusing on stem cells (i.e., why they are biologically important/relevant) and keep that consistency throughout or just generalize the model to all cells. This would help focus the reader’s mind. As well, the authors use the inequalities to describe the scenarios. I think substituting the inequalities with words like positive-frequency dependence, negative-frequency dependence, damaged cells dominant, and intact cells dominant would help with readability. Overall, less wording and being more direct would improve the paper.

The figures for the most part work although it looks like they have been taken from screenshots of a previous paper. This is my biggest concern, that the results have not been already duplicated elsewhere. An explanation of why they look like that would be helpful. I would also modify figure 3 so that there is a second figure (perhaps figure 3b) that shows the difference between theoretical and simulations. One can kind of see the difference in figure 3 but it is quite busy. Having a separate panel that shows just the difference would be helpful especially because in some cases the difference is greater than zero and in other cases it is less than zero. Explaining when we see differences greater than zero versus less than zero and why would just generally be interesting. Perhaps there is a kind of pattern there.

I think the discussion could be better written with the results put in a broader context. I do appreciate discussion on fitting the values to real world data, but there isn’t much beyond that. Reaching into the cancer literature and general insect pest games may help. One potential paper to cite is Kaznatcheev et al. 2019 “Fibroblasts and Alectinib switch the evolutionary games played by non-small cell lung cancer”. In this paper, the authors show that the presence of fibroblasts favor sensitive cancer cells over resistant cancer cells in an evolutionary game, i.e. cell-cell interactions affected the composition and eventual takeover of resistant cancer cells. It seems pertinent to their work.

Did the authors look at all inequalities? I believe there are twelve, ordinally distinct symmetric games. It would be interesting to see how this process affects all twelve types of games.

**Do you want your identity to be public for this peer review?** For information about this choice, including consent withdrawal, please see our Privacy Policy

Reviewer #1: No

Reviewer #2: No

Reviewer #3: No

---

## [Author Response · Author response to Decision Letter 1]

31 Aug 2025

>Reviewer #1: Uchinomiya and Tomita present a mathematical model of the effect of

>competition between “intact” and “damaged” stem cells on the accumulation of damaged cells

>within the population. Interestingly, although the text is focused on low-dose radiation

>exposure, the model itself is not specific to radiation. Overall, I enjoyed reading the

>manuscript, but offer some comments and suggestions below.

[Answer]

We appreciate the positive evaluation of our work by reviewer #1. We understand the concerns of this reviewer and revised the manuscript.

>Major comments

>1. According to the introduction, the goal of this study is to investigate the influence of stem

>cell competition on accumulation of damage from low dose background radiation. However, the

>discussion and conclusion do not clearly or strongly tie the results back to this goal. That is,

>there is no discussion of what implications these results have for protective measures that may

>be taken against low dose rate exposure.

[Answer]

As the reviewer pointed out, there was insufficient discussion. Multiple biological processes are involved in radiation-induced carcinogenesis. In this model, we focused on stem cell competition, a type of cell-cell interaction, and were able to discuss the conditions under which damage accumulation is suppressed by stem cell competition. We believe that future models can be improved to take other biological processes into account, thereby explaining why low-dose-rate exposure does not increase the risk of cancer. We have added a description of this. (Line. 501-506, 530-535)

>2. Lines 380-389. The discussion presented in these lines does not make sense. Please

>completely rewrite this section. Firstly, as far as this reviewer knows and can find, there exists

>no such definition of an “elemental dose” in this context. In addition, reference 8, which is

>given for this definition, does not discuss such a concept. What is the correct reference?

>Second, the calculation that it would take 18 days to reach 1mGy under the assumption of an

>exposure limit of 20mGy per year isn’t valid. The time it takes to accumulate any exposure

>would depend solely on the dose rate being experienced, not the upper limit of dose exposure.

>Third, it is not clear what approximation the authors are referring to in line 387.

[Answer]

The citation was incorrect, so we corrected it to the correct citation. We also made comprehensive corrections to points 2 and 3 that the reviewer pointed out (Line 416-432)。

>3. During model development, several explicit or implicit assumptions were made that need to

>be discussed either in the methods or in the discussion section. For example, it is assumed that

>damage is irreversible. However, it is well known that radiation induced DNA damage is

>repairable and could even occur on the time scales considered in the article.

[Answer]

Although the assumption here is irreversible damage, it can also be interpreted that the probability H includes the effect of repair. We have made this clear in lines 119-120. I have also added more detailed descriptions of some of the assumptions. (Line 496-501)。

>Minor comments

>1. Lines 217-227. The text refers to data presented in Figure 3. There is a mismatch between

>the color of the data in the figure that corresponds to the values (green) and the color used in

>the text (pink).

[Answer]

We revised it according to the comments (Line 243).

>2.Please provide information regarding the initial makeup of the population. Do the

>simulations begin with one damaged cell, or only intact cells?

[Answer]

The fig. 3 starts from only intact cells, while fig. 6 starts from one damaged cell. The description of Fig. 3 was insufficient, so we have clarified it. (Line 224-225)。

>3. The definition of Tabs isn’t clear. Is it when there is only damaged stem cells present?

[Answer]

Tabs is the average waiting time until only damaged stem cells are present. We added a description to clarify this. (Line 175-177)

>4. Can the lattice-based model only support a certain number of cells, or does the lattice

>expand indefinitely?

[Answer]

At least in this model, only a finite number of cells are considered. This is also a limitation of the model related to major's comment 3, so we added a description. (Line 185-186)

>5. How did the authors decide on performing 50 simulations per condition? Currently the first

>mention that 50 simulations were performed is in the caption of Figure 3. Please also mention

>this in the methods section. Further, please adapt the figures (where appropriate) to include

>the standard deviation as error bars.

[Answer]

The results converged to the approximation after 50 simulations. We performed 1,000 simulations and created the graph again. We added “2.4 Analyses of the models” section (Line 215-219). In the current figure, error bars are not shown for clarity.

>6. Please include a brief explanation of how to the C values can be interpreted.

>For example, does C_(I←D)<1 indicate that damaged cells lower or increase

>the cost of intact cells?

[Answer]

We added a description according to the suggestion (Line 178-182)。

>7. The manuscript mentions “low dose-rate” conditions as being when λ is “very small”. In

>Figure 3, λ ranges from 10^-3 to 1. Are all of these values considered to be low dose rate?

[Answer]

In the presented model, the unit of time is one cell cycle, and this must be taken into account when converting to an actual dose rate. However, when λ=1, all cells are damaged more than once on average during the cell cycle, so this cannot be considered a low dose rate.

>8. According to lines 252-254 “…Tabs could be normalized to 1” in Figure 4 (a). Has Tabs

>been normalized to one in the rest of Figure 4, or has it been normalized to the value of Tabs

>from Figure 4(a)? It is not clear.

[Answer]

The Tabs is normalized to 1 in the section. The description has been revised. (Line 279-281)

>9. Lines 377-378 states “For example, the influence of cell competition was significant under

>low dose-rate condition..”. Please clarify if this refers to a statistically significant difference or

>not.

[Answer]

It does not mean statistically significant difference. The description has been corrected Line 413.

>10. Lines 107, 164,230. Please replace the word abstract with abstraction.

[Answer]

We have corrected it according to the suggestion (Lines 98, 141, 207).

>11. Line 109-110 is not clear. Please rephrase.

[Answer]

We have corrected it according to the suggestion (Line 114-115).

>12. Lines 163-164 state “The well-mixed model is a Markov process, and the absorbing state is

>only when damaged cells occupy the cell pool.” Do the authors have a proof for this statement?

>If so, please include it in the appendix or supplemental information.

[Answer]

We added the simple proof in the appendix (Appendix A).

>13. Line 175-176 “Then, one cell is randomly chosen from the removed cell and surrounding

>cells to divide and fill the removed space.” So the removed cell proliferates? Please clarify.

[Answer]

The cell that is eliminated may also proliferate. In the model, the cells that divide and the cells that are eliminated are determined simultaneously. We added a statement for clarity. (Line 192-193)

>14. I encourage the authors to be consistent with the tenses used throughout the manuscript. In

>some places they have used past tense, and in some they use present tense.

[Answer]

We reviewed the manuscript and revised it, especially the 3.Result section.

>15. Lines 255-256 and 258-259. I believe the authors have switched the descriptions. In Figure

>4(b), Tabs monotonically decreases with N, and in Figure 4(c) it monotonically increases with

>N.

[Answer]

We have corrected it according to the suggestion(Line 283-286)。

>16. In line 292, there is a typo. I believe “closed” should be “close”.

[Answer]

We have corrected it according to the suggestion (Line 324).

>17. In line 302, it should be “therefore”.

[Answer]

We have corrected it according to the suggestion (Line 333).

>18. The description of figure 5, and how the dotted and dash dotted lines were chosen and what

>they mean is unclear. Please rephrase the explanation.

[Answer]

The type of line has no particular meaning. It was used to confirm that the numerical calculations were performed correctly. In particular, N needs to be large enough to distinguish between monotonically decreasing and downward convexity. However, this is difficult to check numerically, so this method was used. We added the description (Line 349-352).

>Reviewer #2: Title: A Mathematical Model assuming Frequency-dependent Cost for Analyzing

>the Influence of Cell Competition on Radiation Effects

>Summary: authors use standard Moran process with state-switching model to determine the

>influence of radiation on stem cell damaged cell’s expanding over intact stem cells. Then the

>authors extend the model to a standard spatial Moran process with 8 nearest neighbors.

>Overall the research question is important, but the methods are standard and the connection to

>existing theoretical literature is weak, including any discussion of types of payoff matrices (e.g.

>prisoner’s dilemma; hawk-dove; etc) rather than relying on hard-coded payoff values. There are

>some theoretical analysis of the rate of mutant expansion that might be helpful here, too. For a

>good example, check Heyde et. al. Cell, 2021.

[Answer]

We appreciate the comments of this reviewer #1. We addressed all suggestions. From the viewpoint of game theory, the upper right area of Fig. 5 and S1 Fig is intact cell dominant, and the lower left corner is damaged cell dominant, which corresponds to the prisoner's dilemma (Line 317-318). The lower right shows that interactions with different types of cells have higher cost. This corresponds to coordination game (Line 323). The upper left shows that interacting with different types of cells results in low cost, i.e. high fitness. This parameter sets include hawk-dove game (Line 339).

>The most interesting part of the manuscript is the connection between frequency-dependent

>competition and radiation dose. However the dose parameter, \lambda, is not contained in the

>equations of the competition model in equation 3, making it difficult to determine its influence.

>Secondly, I assume that \lambda will have an effect on stem cell pool size, N, and that radiation

>may influence the competition parameters in the matrix too.

[Answer]

In this model, radiation is assumed to only affect whether cells are damaged or not, and does not affect the competition parameters in the matrix or N. These are limitations of the model and should be considered in future works, so additional descriptions have been added (Line 124-125, 499-501 ).

>One of the central claims of the paper, that “When the dose rate is very low, some stem cells

>may be damaged and mixed with intact stem cells in the stem cell pool” isn’t sufficiently

>explored in the model. For example, an effect of radiation dose is linear, and thus there is no

>qualitative regime change between high and low doses of radiation.

[Answer]

As this comment, this is only a hypothesis in the presented model. It is an issue for future studies, so we have added some details. (Line 496-497)

>Many claim in the introduction or conclusion section are true claims, but do not have

>associated references; for example:

>1. Line 36: “Ionizing radiation is a well-known external factor that increases the risk of cancer. “

>2. Line 59: “The hypothesis that stem cell competition suppresses the effects of radiation has

>been supported by experiments.”

>3. Line 462: “In recent years, the effects of cell-cell interactions have begun to be measured

>quantitatively.”

[Answer]

Following the suggestions, we added references corresponds to suggestions 1 and 2. Regarding suggestion 3, [17-19] corresponds to the references, but we did not add it because it is the conclusion section.

>Reviewer #3: I have read the article “A Mathematical Model assuming Frequency-dependent

>Cost for Analyzing the Influence of Cell Competition on Radiation Effects.” In this manuscript,

>the authors analyze the effect of cell-cell competition on the accumulation of cells damaged by

>radiation. They constructed mathematical model of radiation damage which used a Moran

>process to model growth and death of cells, a Markov model for transition probabilities

>between induced and damaged cells, and an evolutionary game of interaction between cells.

>They analyzed this under a well-mixed and spatially explicit model. They showed that

>frequency-dependent competition between cells can slow down the rate of damaged cell

>takeover, at least under low-dose radiation. Overall, this is a good paper with an appropriate

>model that shows interesting results. I cannot find serious flaws with the methodology or

>analysis. I list my comments regarding the manuscript below.

We appreciate the positive comments of reviewer #3. Each comment has been considered, and we have revised the manuscript accordingly in response to all points.

>My first comment is that elements of the manuscript could be made more clearly. Firstly, the

>authors mention cell fitness. It would be good to define fitness in the context of cells. I have an

>understanding of what they mean but a formal definition would help.

[Answer]

Fitness represents the proliferation ability of a cell and is affected by cell division, death, etc. Definition has been added (Line 55-58, 108-110).

>They also mention stem cell competition but it seems like this model could be broadened to any

>kind of cell. I would either state the reasons for focusing on stem cells (i.e., why they are

>biologically important/relevant) and keep that consistency throughout or just generalize the

>model to all cells. This would help focus the reader’s mind.

[Answer]

Stem cells are the target of radiation induced cancer, so we focused on stem cells here. However, the model itself can also be applied to other cells. We added a description (Line 410-412).

>As well, the authors use the inequalities to describe the scenarios. I think substituting the

>inequalities with words like positive-frequency dependence, negative-frequency dependence,

>damaged cells dominant, and intact cells dominant would help with readability. Overall, less

>wording and being more direct would improve the paper.

[Answer]

We have introduced those terms as suggested.

>The figures for the most part work although it looks like they have been taken from screenshots

>of a previous paper. This is my biggest concern, that the results have not been already

>duplicated elsewhere. An explanation of why they look like that would be helpful. I would also

>modify figure 3 so that there is a second figure (perhaps figure 3b) that shows the difference

>between theoretical and simulations. One can kind of see the difference in figure 3 but it is

>quite busy. Having a separate panel that shows just the difference would be helpful especially

>because in some cases the difference is greater than zero and in other cases it is less than zero.

>Explaining when we see differences greater than zero versus less than zero and why would just

>generally be interesting. Perhaps there is a kind of pattern there.

[Answer]

The figure is completely original, but I think there are some issues with the image quality being limited during the updating process. We update figures again. As the absolute values are different, the difference between the simulation and approximation seen in Figure 3a (original Figure 3) is difficult to see by taking the difference. We have added the ratio of the simulation to the approximation as Figure 3b.

>I think the discussion could be better written with the results put in a broader context. I do

>appreciate discussion on fitting the values to real world data, but there isn’t much beyond that.

>Reaching into the cancer literature and general insect pest games may help. One potential

>paper to cite is Kaznatcheev et al. 2019 “Fibroblasts a

---

## [Decision Letter · Decision Letter 1]

18 Sep 2025

Dear Dr. Uchinomiya,

Thank you for submitting your manuscript to PLOS ONE. After careful consideration, we feel that it has merit but does not fully meet PLOS ONE’s publication criteria as it currently stands. Therefore, we invite you to submit a revised version of the manuscript that addresses the points raised during the review process.

We look forward to receiving your revised manuscript.

Kind regards,

Attila Csikász-Nagy

Academic Editor

PLOS ONE

Journal Requirements:

**Additional Editor Comments:**

Address the remaining concerns of reviewer 2.

Reviewers' comments:

Reviewer's Responses to Questions

**Comments to the Author**

Reviewer #1: All comments have been addressed

Reviewer #2: (No Response)

Reviewer #3: All comments have been addressed

2. Is the manuscript technically sound, and do the data support the conclusions?

Reviewer #1: Yes

Reviewer #2: Partly

Reviewer #3: Yes

3. Has the statistical analysis been performed appropriately and rigorously?

Reviewer #1: N/A

Reviewer #2: N/A

Reviewer #3: Yes

4. Have the authors made all data underlying the findings in their manuscript fully available?

Reviewer #1: No

Reviewer #2: Yes

Reviewer #3: Yes

5. Is the manuscript presented in an intelligible fashion and written in standard English?

Reviewer #1: Yes

Reviewer #2: Yes

Reviewer #3: Yes

Reviewer #1: (No Response)

Reviewer #2: My previous concern that “the dose parameter, \lambda, is not contained in the equations of the competition model in equation 3, making it difficult to determine its influence” was not addressed. I remain concerned that the model methods still isn't written in such a way that is reproducible.

Based on the authors' response, i now understand that the Moran process has 3 steps: transition, birth, replacement. The equations in 3a, 3b, 3c, and 3d only describe birth and replacement. Please consider updating these equations to be explicit functions of lambda, to include the transition step.

All my other comments have been addressed satisfactorily.

Reviewer #3: The reviewers have taken all my concerns into account, and I am pleased with the state of the manuscript.

**Do you want your identity to be public for this peer review?** For information about this choice, including consent withdrawal, please see our Privacy Policy

Reviewer #1: No

Reviewer #2: No

Reviewer #3: No

---

## [Author Response · Author response to Decision Letter 2]

30 Oct 2025

Response to Reviewer #2:

>My previous concern that “the dose parameter, \lambda, is not contained in the equations of the

>competition model in equation 3, making it difficult to determine its influence” was not addressed. I

>remain concerned that the model methods still isn't written in such a way that is reproducible.

>Based on the authors' response, i now understand that the Moran process has 3 steps: transition, birth,

>replacement. The equations in 3a, 3b, 3c, and 3d only describe birth and replacement. Please consider

>updating these equations to be explicit functions of lambda, to include the transition step.

[Answer]

The dose rate λ can affect the equations 3a, 3b, 3c, and 3d through cost parameters. However, detailed analyses of this assumption would be complex and be beyond the scope of the present study, so we did not modify the equations. Nevertheless, the reviewer’s comment is biologically important point. We have therefore included a discussion in the Discussion section on how the properties of damaged cells may relate to the λ-dependence of the parameters (Line 501-508).

---

## [Editor Report · Decision Letter 2]

5 Nov 2025

A Mathematical Model assuming Frequency-dependent Cost for Analyzing the Influence of Cell Competition on Radiation Effects

PONE-D-25-19152R2

Dear Dr. Uchinomiya,

We’re pleased to inform you that your manuscript has been judged scientifically suitable for publication and will be formally accepted for publication once it meets all outstanding technical requirements.

Kind regards,

Attila Csikász-Nagy

Academic Editor

PLOS ONE
---

## [Editor Report · Acceptance letter]

PONE-D-25-19152R2

PLOS ONE

Dear Dr. Uchinomiya,

I'm pleased to inform you that your manuscript has been deemed suitable for publication in PLOS ONE. Congratulations! Your manuscript is now being handed over to our production team.

Kind regards,

on behalf of

Dr. Attila Csikász-Nagy

Academic Editor

PLOS ONE